# Variational Information Bottleneck for Effective Low-Resource Fine-Tuning

**Rabeeh Karimi Mahabadi**♣♡ **Yonatan Belinkov**◇* **James Henderson**♣

♡EPFL, Switzerland

♣Idiap Research Institute, Switzerland

◇Technion – Israel Institute of Technology

{rabeeh.karimi,james.henderson}@idiap.ch
belinkov@technion.ac.il

## Abstract

While large-scale pretrained language models have obtained impressive results when fine-tuned on a wide variety of tasks, they still often suffer from overfitting in low-resource scenarios. Since such models are general-purpose feature extractors, many of these features are inevitably irrelevant for a given target task. We propose to use Variational Information Bottleneck (VIB) to suppress irrelevant features when fine-tuning on low-resource target tasks, and show that our method successfully reduces overfitting. Moreover, we show that our VIB model finds sentence representations that are more robust to biases in natural language inference datasets, and thereby obtains better generalization to out-of-domain datasets. Evaluation on seven low-resource datasets in different tasks shows that our method significantly improves transfer learning in low-resource scenarios, surpassing prior work. Moreover, it improves generalization on 13 out of 15 out-of-domain natural language inference benchmarks. Our code is publicly available in `https://github.com/rabeehk/vibert`.

## 1 Introduction

Transfer learning has emerged as the de facto standard technique in natural language processing (NLP), where large-scale language models are pretrained on an immense amount of text to learn a general-purpose representation, which is then transferred to the target domain with fine-tuning on target task data. This method has exhibited state-of-the-art results on a wide range of NLP benchmarks (Devlin et al., 2019; Liu et al., 2019; Radford et al., 2019). However, such pretrained models have a huge number of parameters, potentially making fine-tuning susceptible to overfitting.

In particular, the task-universal nature of large-scale pretrained sentence representations means that much of the information in these representations is irrelevant to a given target task. If the amount of target task data is small, it can be hard for fine-tuning to distinguish relevant from irrelevant information, leading to overfitting on statistically spurious correlations between the irrelevant information and target labels. Learning low-resource tasks is an important topic in NLP (Cherry et al., 2019) because annotating more data can be very costly and time-consuming, and because in several tasks access to data is limited.

In this paper, we propose to use the Information Bottleneck (IB) principle (Tishby et al., 1999) to address this problem of overfitting. More specifically, we propose a fine-tuning method that uses Variational Information Bottleneck (VIB; Alemi et al. 2017) to improve transfer learning in low-resource scenarios.

VIB addresses the problem of overfitting by adding a regularization term to the training loss that directly suppresses irrelevant information. As illustrated in Figure 1, the VIB component maps the sentence embedding from the pretrained model to a latent representation $z$, which is the only input to the task-specific classifier. The information that is represented in $z$ is chosen based on the IB principle, namely that all the information about the input that is represented in $z$ should be necessary for the task. In particular, VIB directly tries to remove the irrelevant information, making it easier for the task classifier to avoid overfitting when trained on a small amount of data. We find that in low-resource scenarios, using VIB to suppress irrelevant features in pretrained sentence representations substantially improves accuracy on the target task.

---

*Supported by the Viterbi Fellowship in the Center for Computer Engineering at the Technion.

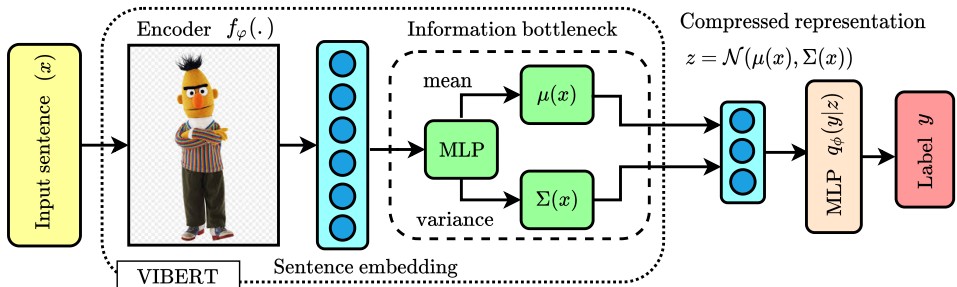

Figure 1: VIBERT compresses the encoder's sentence representation $f_\varphi(x)$ into representation $z$ with mean $\mu(x)$ and eliminates irrelevant and redundant information through the Gaussian noise with variance $\Sigma(x)$.

Removing unnecessary information from the sentence representation also implies removing redundant information. VIB tries to find the most concise representation which can still solve the task, so even if a feature is useful alone, it may be removed if it isn't useful when added to other features because it is redundant. We hypothesize that this provides a useful inductive bias for some tasks, resulting in better generalization to out-of-domain data. In particular, it has recently been demonstrated that annotation biases and artifacts in several natural language understanding benchmarks (Kaushik & Lipton, 2018; Gururangan et al., 2018; Poliak et al., 2018; Schuster et al., 2019) allow models to exploit superficial shortcuts during training to perform surprisingly well without learning the underlying task. However, models that rely on such superficial features do not generalize well to out-of-domain datasets, which do not share the same shortcuts (Belinkov et al., 2019a). We investigate whether using VIB to suppress redundant features in pretrained sentence embeddings has the effect of removing these superficial shortcuts and keeping the deep semantic features that are truly useful for learning the underlying task. We find that using VIB does reduce the model's dependence on shortcut features and substantially improves generalization to out-of-domain datasets.

We evaluate the effectiveness of our method on fine-tuning BERT (Devlin et al., 2019), which we call the VIBERT model (Variational Information Bottleneck for Effective Low-Resource Fine-Tuning). On seven different datasets for text classification, natural language inference, similarity, and paraphrase tasks, VIBERT shows greater robustness to overfitting than conventional fine-tuning and other regularization techniques, improving accuracies on low-resource datasets. Moreover, on NLI datasets, VIBERT shows robustness to dataset biases, obtaining substantially better generalization to out-of-domain NLI datasets. Further analysis demonstrates that VIB regularization results in less biased representations. Our approach is highly effective and simple to implement, involving a small additional MLP classifier on top of the sentence embeddings. It is model agnostic and end-to-end trainable.

In summary, we make the following contributions: 1) Proposing VIB for low-resource fine-tuning of large pretrained language models. 2) Showing empirically that VIB reduces overfitting, resulting in substantially improved accuracies on seven low-resource benchmark datasets against conventional fine-tuning and prior regularization techniques. 3) Showing empirically that training with VIB is more robust to dataset biases in NLI, resulting in significantly improved generalization to out-of-domain NLI datasets. To facilitate future work, we will release our code.

## 2 FINE-TUNING IN LOW-RESOURCE SETTINGS

The standard fine-tuning paradigm starts with a large-scale pretrained model such as BERT, adds a task-specific output component which uses the pretrained model's sentence representation, and trains this model end-to-end on the task data, fine-tuning the parameters of the pretrained model. As depicted in Figure 1, we propose to add a VIB component that controls the flow of information from the representations of the pretrained model to the output component. The goal is to address overfitting in resource-limited scenarios by removing irrelevant and redundant information from the pretrained representation.

**Problem Formulation**   We consider a general multi-class classification problem with a low-resource dataset $\mathcal{D} = \{x_i, y_i\}_{i=1}^{N}$ consisting of inputs $x_i \in \mathcal{X}$, and labels $y_i \in \mathcal{Y}$. We assume we are also given a large-scale pretrained encoder $f_\varphi(.)$ parameterized by $\varphi$ that computes sentence embeddings for the input $x_i$. Our goal is to fine-tune $f_\varphi(.)$ on $\mathcal{D}$ to maximize generalization.

**Information Bottleneck**   To specifically optimize for the removal of irrelevant and redundant information from the input representations, we adopt the Information Bottleneck principle. The objective of IB is to find a maximally compressed representation $Z$ of the input representation $X$ (compression loss) that

maximally preserves information about the output $Y$ (prediction loss),[1] by minimizing:

$$\mathcal{L}_{\text{IB}} = \underbrace{\beta I(X, Z)}_{\text{Compression Loss}} - \underbrace{I(Z, Y)}_{\text{Prediction Loss}}, \tag{1}$$

where $\beta \geq 0$ controls the balance between compression and prediction, and $I(.,.)$ is the mutual information.

**Variational Information Bottleneck**    Alemi et al. (2017) derive an efficient variational estimate of (1):

$$\mathcal{L}_{\text{VIB}} = \beta \, \mathbb{E}_{x}[\text{KL}[p_{\theta}(z|x), r(z)]] + \mathbb{E}_{z \sim p_{\theta}(z|x)}[-\log q_{\phi}(y|z)], \tag{2}$$

where $q_{\phi}(y|z)$ is a parametric approximation of $p(y|z)$, $r(z)$ is an estimate of the prior probability $p(z)$ of $z$, and $p_{\theta}(z|x)$ is an estimate of the posterior probability of $z$. During training, the compressed sentence representation $z$ is sampled from the distribution $p_{\theta}(z|x)$, meaning that a specific pattern of noise is added to the input of the output classifier $q_{\phi}(y|z)$. Increasing this noise decreases the information conveyed by $z$. In this way, the VIB module can block the output classifier $q_{\phi}(y|z)$ from learning to use specific information. At test time, the expected value of $z$ is used for predicting labels with $q_{\phi}(y|z)$. We refer to the dimensionality of $z$ as $K$, which specifies the bottleneck size. Note that there is an interaction between decreasing $K$ and increasing the compression by increasing $\beta$ (Shamir et al., 2010; Harremoës & Tishby, 2007). $K$ and $\beta$ are hyper-parameters (Alemi et al., 2017).

We consider parametric Gaussian distributions for prior $r(z)$ and $p_{\theta}(z|x)$ to allow an analytic computation for their Kullback-Leibler divergence,[2] namely $r(z) = \mathcal{N}(z|\mu_0, \Sigma_0)$ and $p_{\theta}(z|x) = \mathcal{N}(z|\mu(x), \Sigma(x))$, where $\mu$ and $\mu_0$ are $K-$dimensional mean vectors, and $\Sigma$ and $\Sigma_0$ are diagonal covariance matrices. We use the reparameterization trick (Kingma & Welling, 2013) to estimate the gradients, namely $z = \mu(x) + \Sigma(x) \odot \epsilon$, where $\epsilon \sim \mathcal{N}(0, I)$. To compute the compressed sentence representations $p_{\theta}(z|x)$, as shown in Figure 1, we first feed sentence embeddings $f_{\varphi}(x)$ through a shallow MLP. It is then followed by two linear layers, each with $K$ hidden units to compute $\mu(x)$ and $\Sigma(x)$ (after a softplus transform to ensure non-negativity). We also use another linear layer to approximate $q_{\phi}(y|z)$.

## 3    EXPERIMENTS

**Datasets**    We evaluate the performance on seven different benchmarks for multiple tasks, in particular text classification, natural language inference, similarity, and paraphrase detection. For NLI, we experiment with two well-known NLI benchmarks, namely SNLI (Bowman et al., 2015) and MNLI (Williams et al., 2018). For text classification, we evaluate on two sentiment analysis datasets, namely IMDB (Maas et al., 2011) and Yelp2013 (YELP) (Zhang et al., 2015). We additionally evaluate on three low-resource datasets in the GLUE benchmark (Wang et al., 2019):[3] paraphrase detection using MRPC (Dolan & Brockett, 2005), semantic textual similarity using STS-B (Cer et al., 2017), and textual entailment using RTE (Dagan et al., 2006). For the GLUE benchmark, SNLI, and Yelp, we evaluate on the standard validation and test splits. For MNLI, since the test sets are not available, we tune on the matched dev set and evaluate on the mismatched dev set (MNLI-M) or vice versa. See Appendix A for datasets statistics and Appendix B for hyper-parameters of all methods.

**Base Model**    We use the BERT$_{\text{Base}}$ (12 layers, 110M parameters) and BERT$_{\text{Large}}$ (24 layers, 340M parameters) uncased (Devlin et al., 2019) implementation of Wolf et al. (2019) as our base models,[4] known to work well for these tasks. We use the default hyper-parameters of BERT, i.e., we use a sequence length of 128, with batch size 32. We use the stable variant of the Adam optimizer (Zhang et al., 2021; Mosbach et al., 2021) with the default learning rate of $2\text{e}-5$ through all experiments. We do not use warm-up or weight decay.

**Baselines**    We compare against prior regularization techniques, including previous state-of-the-art, Mixout:

---

[1] In this work, $Z$, $X$, and $Y$ are random variables, and $z$, $x$ and $y$ are instances of these random variables.

[2] $\text{KL}(\mathcal{N}(\mu_0, \Sigma_0) \| \mathcal{N}(\mu_1, \Sigma_1)) = \frac{1}{2}(\text{tr}(\Sigma_1^{-1}\Sigma_0) + (\mu_1 - \mu_0)^T \Sigma_1^{-1}(\mu_1 - \mu_0) - K + \log(\frac{\det(\Sigma_1)}{\det(\Sigma_0)}))$.

[3] We did not evaluate on WNLI and CoLA due to the irregularities in these datasets and the reported instability during the fine-tuning https://gluebenchmark.com/faq.

[4] To have a controlled comparison, all results are computed with this PyTorch implementation, which might slightly differ from the TensorFlow variant (Devlin et al., 2019).

- **Dropout** (Srivastava et al., 2014), a widely used stochastic regularization techniques used in multiple large-scale language models (Devlin et al., 2019; Yang et al., 2019; Vaswani et al., 2017) to mitigate overfitting. Following Devlin et al. (2019), we apply dropout on all layers of BERT.

- **Mixout** (Lee et al., 2019) is a stochastic regularization technique inspired by Dropout with the goal of preventing catastrophic forgetting during fine-tuning. Mixout regularizes the learning to minimize the deviation of a fine-tuned model from the pretrained initialization. It replaces the model parameters with the corresponding value from the pretrained model with probability $p$.

- **Weight Decay (WD)** is a common regularization technique to improve generalization (Krogh & Hertz, 1992). It regularizes the large weights $w$ by adding a penalization term $\frac{\lambda}{2}\|w\|$ to the loss, where $\lambda$ is a hyperparameter specifying the strength of regularization. Chelba & Acero (2004) and Daumé III (2007) adapt WD for fine-tuning of the pretrained models, and propose to replace this regularization term with $\lambda\|w - w_0\|$, where $w_0$ are the weights of the pretrained models. Recently, Lee et al. (2019) demonstrated that the latter formulation of WD works better for fine-tuning of BERT than conventional WD and can improve generalization on small training sets.

## 3.1 RESULTS ON THE GLUE BENCHMARK

Table 1 shows results on the low-resource datasets in GLUE.[5] We find that a) Our VIBERT model substantially outperforms the baselines on all the datasets, demonstrating the effectiveness of the proposed method. b) Dropout decreases the performance on low-resource datasets. We conjecture that regularization techniques relying on stochasticity without considering the relevance to the output, in contrast to VIB, can make it more difficult for learning to extract relevant information from a small amount of data. Igl et al. (2019) observe similar effects in another application. c) Similar to the results of Zhang et al. (2021), we find less pronounced benefits of the previously suggested methods than the results originally published. This can be explained by using a more stable version of Adam (Zhang et al., 2021) suggested by the very recent work in our experiments, which decreases the added benefits of previously suggested regularization techniques on top of a stable optimizer. In contrast, our VIBERT model still substantially improves the results and surpasses the prior work in all settings for both $\text{BERT}_{\text{Base}}$ and $\text{BERT}_{\text{Large}}$ models. Due to the computational overhead of $\text{BERT}_{\text{Large}}$, for the rest of this work, we stick to $\text{BERT}_{\text{Base}}$.

Table 1: Average results and standard deviation in parentheses over 3 runs on low-resource data in GLUE. $\mathbf{\Delta}$ shows the absolute difference between the results of the VIBERT model with BERT.

| | MRPC | | STS-B | | RTE |
|---|---|---|---|---|---|
| **Model** | **Accuracy** | **F1** | **Pearson** | **Spearman** | **Accuracy** |
| $\text{BERT}_{\text{Base}}$ | 87.80 (0.5) | 83.20 (0.6) | 84.93 (0.1) | 83.53 (0.0) | 67.93 (1.5) |
| +Dropout (Srivastava et al., 2014) | 87.33 (0.2) | 81.90 (0.7) | 84.33 (0.9) | 82.73(1.0) | 65.80 (1.5) |
| +Mixout (Lee et al., 2019) | 87.03 (0.2) | 82.63 (0.3) | 85.23 (0.4) | 83.80(0.4) | 67.70 (0.9) |
| +WD (Lee et al., 2019) | 87.57(0.2) | 82.83(0.3) | 85.0(0.3) | 83.6(0.2) | 68.63(1.3) |
| $\text{VIBERT}_{\text{Base}}$ | **89.23 (0.1)** | **85.23 (0.2)** | **87.63 (0.3)** | **86.50 (0.4)** | **70.53 (0.5)** |
| $\mathbf{\Delta}$ | +1.43 | +2.03 | +2.7 | +2.97 | +2.6 |
| $\text{BERT}_{\text{Large}}$ | 88.47 (0.7) | 84.20 (1.3) | 86.87 (0.2) | 85.70 (0.1) | 68.67 (0.8) |
| +Dropout (Srivastava et al., 2014) | 87.77 (0.4) | 82.97 (0.2) | 86.47 (0.1) | 85.33 (0.2) | 65.77 (0.6) |
| +Mixout (Lee et al., 2019) | 88.57 (0.7) | 84.10 (1.1) | 86.70 (0.2) | 85.43 (0.3) | 70.03 (1.0) |
| +WD (Lee et al., 2019) | 88.97(0.5) | 84.87(0.4) | 86.9(0.1) | 85.67(0.1) | 69.27(0.9) |
| $\text{VIBERT}_{\text{Large}}$ | **89.10 (0.4)** | **85.13 (0.6)** | **87.53 (0.8)** | **86.40 (0.9)** | **71.37 (0.8)** |
| $\mathbf{\Delta}$ | +0.63 | +0.93 | +0.66 | +0.7 | +2.7 |

**Impact of Random Seeds:** Following Dodge et al. (2020), we examine the choice of random seed and evaluate the performance of VIBERT and BERT by fine-tuning them across 50 random seeds on GLUE. To comply with the limited access to the GLUE benchmark online system, we split the original validation sets into half and consider one half as the validation set and use the other half as the test set. We first perform model selection on the validation set to fix the hyper-parameters and then fine-tune the selected models for 50 different seeds. Figure 2 shows the expected test performance (Dodge et al., 2019) as the function

---

[5]Note that the test sets are not publicly available and the prior work reports the results on the validation set of the GLUE benchmark (Lee et al., 2019; Dodge et al., 2020). We, however, report the results of their methods and ours on the original test sets by submitting to an online system.

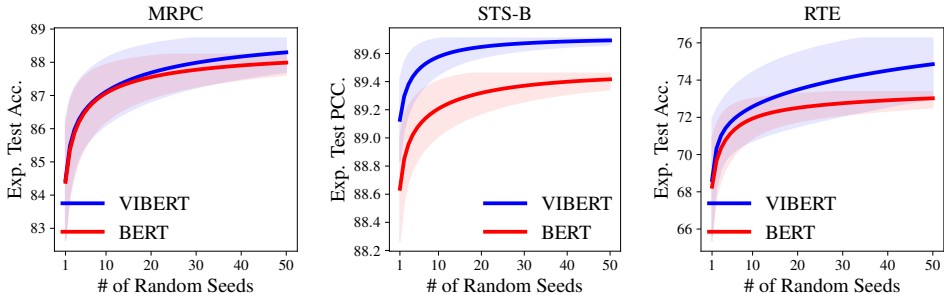

Figure 2: Expected test performance (solid lines) with standard deviation (shaded region) over the number of random seeds allocated for fine-tuning. Our VIBERT model consistently outperforms BERT. We report the accuracy for RTE and MRPC and the Pearson correlation coefficient for STS-B.

of random trials. The results demonstrate that our VIBERT model consistently obtains better performance than BERT on all datasets. As anticipated, the expected test performance monotonically increases with more random trials (Dodge et al., 2020) till it reaches a plateau, such as after 30 trials on STS-B.

## 3.2 VARYING-RESOURCE RESULTS

To analyze the performance of our method as a function of dataset size, we use four large-resource NLI and sentiment analysis datasets, namely SNLI, MNLI, IMDB, and YELP to be able to subsample the training data with varying sizes. Table 2 shows the obtained results. VIBERT consistently outperforms all the baselines on low-resource scenarios, but the advantages are reduced or eliminated as we approach a medium-resource scenario. Also, the improvements are generally larger when the datasets are smaller, showing that our method successfully addresses low-resource scenarios.

Table 2: Test accuracies in the low-resource setting on text classification and NLI datasets under varying sizes of training data (200, 500, 800, 1000, 3000, and 6000 samples). We report the average and standard deviation in parentheses across three runs. We show the highest average result in each setting in bold. Δ shows the absolute difference between the results of VIBERT with BERT.

| Data | Model | 200 | 500 | 800 | 1000 | 3000 | 6000 |
|------|-------|-----|-----|-----|------|------|------|
| SNLI | BERT | 58.70 (1.3) | 68.12 (1.5) | 73.29 (0.9) | 74.69 (1.1) | 79.57 (0.4) | 80.85 (0.4) |
| | +Dropout | 58.95 (0.4) | 69.33 (1.1) | 73.22 (1.2) | 74.20 (0.5) | 79.48 (0.7) | **81.71** (0.6) |
| | +Mixout | 58.52 (1.3) | 68.26 (1.7) | 72.81 (1.0) | 74.09 (0.5) | 78.7 (0.3) | 80.61 (0.5) |
| | +WD | 59.23 (1.5) | 68.54 (1.9) | 73.72 (1.0) | 74.78 (0.8) | **79.83** (0.5) | 81.32 (0.5) |
| | VIBERT | **61.42** (1.3) | **70.75** (0.6) | **74.71** (0.5) | **75.84** (0.1) | 79.56 (0.3) | 81.29 (0.4) |
| | Δ | +2.72 | +2.63 | +1.42 | +1.15 | -0.01 | +0.44 |
| MNLI | BERT | 49.93 (1.4) | 59.76 (2.0) | 63.63 (1.6) | 65.21 (1.4) | 70.67 (0.7) | 73.11 (0.9) |
| | +Dropout | 50.74 (2.1) | 59.58 (2.1) | 62.82 (0.8) | 65.71 (1.4) | 71.11 (0.8) | 72.88 (1.1) |
| | +Mixout | 50.05 (1.8) | 58.69 (2.8) | 63.31 (1.7) | 64.58 (1.5) | 70.60 (0.8) | 72.56 (0.7) |
| | +WD | 49.92 (1.4) | 60.36 (2.0) | 64.41 (1.5) | 65.3 (1.0) | 71.47 (0.8) | 72.94 (0.7) |
| | VIBERT | **53.58** (0.9) | **63.04** (1.1) | **64.87** (0.6) | **66.41** (1.2) | **71.86** (0.9) | **74.22** (0.3) |
| | Δ | +3.65 | +3.28 | +1.24 | +1.2 | +1.19 | +1.11 |
| IMDB | BERT | 78.96 (1.9) | 83.68 (0.2) | 84.04 (0.9) | 84.80 (0.0) | 86.17 (0.2) | 86.98 (0.4) |
| | +Dropout | 81.19 (1.6) | 83.30 (0.2) | 84.52 (0.3) | 85.01 (0.3) | 86.20 (0.2) | **87.31** (0.2) |
| | +Mixout | 79.17 (4.2) | 83.55 (0.3) | 84.37 (0.3) | 84.50 (0.1) | 86.15 (0.1) | 86.97 (0.1) |
| | +WD | 79.78 (2.2) | 83.95 (0.2) | 84.29 (0.6) | 84.97 (0.2) | 86.13 (0.3) | 87.2 (0.1) |
| | VIBERT | **83.05** (0.3) | **84.46** (0.4) | **84.83** (0.4) | **85.03** (0.4) | **86.27** (0.4) | 87.15 (0.3) |
| | Δ | +4.09 | +0.78 | +0.79 | +0.23 | +0.1 | +0.17 |
| YELP | BERT | 41.60 (0.9) | 44.12 (1.4) | 45.67 (1.6) | 46.77 (0.5) | 50.14 (0.7) | 51.86 (0.4) |
| | +Dropout | 41.30 (0.3) | 44.37 (0.6) | 46.49 (0.8) | 46.21 (1.5) | **51.09** (0.2) | **52.39** (0.5) |
| | +Mixout | 41.52 (0.9) | 43.60 (1.1) | 45.65 (1.9) | 46.98 (1.1) | 50.68 (0.5) | 51.51 (0.3) |
| | +WD | 41.66 (0.6) | 44.43 (1.2) | 46.26 (1.4) | 47.37 (0.6) | 50.7 (0.5) | 51.9 (0.6) |
| | VIBERT | **42.30** (0.2) | **46.65** (0.5) | **46.60** (0.1) | **48.03** (0.6) | 50.37 (0.4) | 51.34 (0.4) |
| | Δ | +0.7 | +2.53 | +0.93 | +1.26 | +0.23 | -0.52 |

### 3.3 OUT-OF-DOMAIN GENERALIZATION

Besides improving fine-tuning on low-resource data by removing irrelevant features, we expect VIB to improve on out-of-domain data because it removes redundant features. In particular, annotation artifacts create shortcut features, which are superficial cues correlated with a label (Gururangan et al., 2018; Poliak et al., 2018) that do not generalize well to out-of-domain datasets (Belinkov et al., 2019a). Since solving the real underlying task can be done without these superficial shortcuts, they must be redundant with the deep semantic features that are truly needed. We hypothesize that many more superficial shortcut features are needed to reach the same level of performance as a few deep semantic features. If so, then VIB should prefer to keep the concise deep features and remove the abundant superficial features, thus encouraging the classifier to rely on the deep semantic features, and therefore resulting in better generalization to out-of-domain data. To evaluate out-of-domain generalization, we take NLI models trained on medium-sized 6K subsampled SNLI and MNLI in Section 3.2 and evaluate their generalization on several NLI datasets.

**Datasets:** We consider a total of 15 different NLI datasets used in Mahabadi et al. (2020), including SICK (Marelli et al., 2014), ADD1 (Pavlick & Callison-Burch, 2016), JOCI (Zhang et al., 2017), MPE (Lai et al., 2017), MNLI, SNLI, SciTail (Khot et al., 2018), and three datasets from White et al. (2017) namely DPR (Rahman & Ng, 2012), FN+ (Pavlick et al., 2015), SPR (Reisinger et al., 2015), and Quora Question Pairs (QQP) interpreted as an NLI task as by Gong et al. (2017). We use the same split used in Wang et al. (2017). We also consider SNLI hard and MNLI(-M) Hard sets (Gururangan et al., 2018), a subset of SNLI/MNLI(-M) where a hypothesis-only model cannot correctly predict the labels and the known biases are avoided. Since the target datasets have different label spaces, during the evaluation, we map predictions to each target dataset's space (Appendix C). Following prior work (Belinkov et al., 2019a; Mahabadi et al., 2020), we select hyper-parameters based on the development set of each target dataset and report the results on the test set.

**Results:** Table 3 shows the results of VIBERT and BERT. We additionally include WD, the baseline that performed the best on average on SNLI and MNLI in Table 2. On models trained on SNLI, VIBERT improves the transfer on 13 out of 15 datasets, obtaining a substantial average improvement of 5.51 points. The amount of improvement on different datasets varies, with the largest improvement on SPR and SciTail with +15.5, and +12.5 points respectively, while WD on average obtains only 0.99 points improvement. On models trained on MNLI, VIBERT improves the transfer on 13 datasets, obtaining an average improvement of 3.83 points. The improvement varies across the datasets, with the largest on ADD1 and JOCI with 16.8 and 8.3 points respectively, substantially surpassing WD. Interestingly, VIBERT improves the results on the SNLI and MNLI(-M) hard sets, resulting in models that are more robust to known biases. These results support our claim that VIBERT motivates learning more general features, rather than redundant superficial features, leading to an improved generalization to datasets without these superficial biases. In the next section, we analyze this phenomenon more.

Table 3: Test accuracy of models transferring to new target datasets. All models are trained on SNLI or MNLI and tested on the target datasets. $\Delta$ are absolute differences with BERT.

| Data | SNLI | | | | | MNLI | | | | |
|------|------|--------|-------|-------|-------|------|--------|-------|-------|-------|
| | BERT | VIBERT | $\Delta$ | WD | $\Delta$ | BERT | VIBERT | $\Delta$ | WD | $\Delta$ |
| SICK | 48.47 | 54.68 | +6.2 | 48.37 | -0.1 | 59.16 | 69.17 | +10.0 | 63.87 | +4.7 |
| ADD1 | 78.81 | 84.75 | +5.9 | 80.62 | +1.8 | 66.15 | 82.95 | +16.8 | 67.18 | +1.0 |
| DPR | 50.78 | 50.14 | -0.6 | 50.41 | -0.4 | 49.95 | 49.95 | 0.0 | 49.95 | 0. |
| SPR | 50.21 | 65.68 | +15.5 | 51.90 | +1.7 | 59.16 | 65.61 | +6.5 | 57.21 | -1.9 |
| FN+ | 50.78 | 53.44 | +2.7 | 50.58 | -0.2 | 46.28 | 49.94 | +3.7 | 46.34 | +0.1 |
| JOCI | 42.03 | 50.66 | +8.6 | 43.91 | +1.9 | 45.60 | 53.94 | +8.3 | 46.49 | +0.9 |
| MPE | 58.30 | 58.10 | -0.2 | 58.10 | -0.2 | 55.10 | 50.30 | -4.8 | 58.2 | +3.1 |
| SCITAIL | 62.32 | 74.84 | +12.5 | 65.10 | +2.8 | 72.58 | 75.68 | +3.1 | 75.73 | +3.2 |
| QQP | 65.19 | 70.67 | +5.5 | 65.90 | +0.7 | 67.88 | 70.50 | +2.6 | 68.75 | +0.9 |
| SNLI Hard | 65.72 | 68.35 | +2.6 | 66.82 | +1.1 | 56.98 | 60.29 | +3.3 | 57.8 | +0.8 |
| MNLI Hard | 46.31 | 53.17 | +6.9 | 47.42 | +1.1 | 59.74 | 61.19 | +1.4 | 60.08 | +0.3 |
| MNLI-M Hard | 46.12 | 52.38 | +6.3 | 46.82 | +0.7 | 60.55 | 61.03 | +0.5 | 59.77 | -0.8 |
| SNLI | 80.54 | 81.81 | +1.3 | 81.26 | +0.7 | 64.32 | 67.87 | +3.6 | 65.44 | +1.1 |
| MNLI-M | 60.51 | 64.88 | +4.4 | 62.11 | +1.6 | 72.42 | 73.06 | +0.6 | 72.76 | +0.3 |
| MNLI | 61.79 | 66.76 | +5.0 | 63.42 | +1.6 | 72.73 | 74.67 | +1.9 | 72.89 | +0.2 |
| Average | — | — | +5.51 | — | +0.99 | — | — | +3.83 | — | +0.93 |

## 4 ANALYSIS

**Analysis of the Removed Features**   Elazar & Goldberg (2018) propose a challenging framework to evaluate if debiasing methods have succeeded in removing biases from the sentence representation. After debiasing, the trained encoder is frozen and the classifier is retrained to try to extract the biases. If the classifier reaches high accuracy given only bias features, then the encoder's representation has not been successfully debiased. We follow the framework of Elazar & Goldberg (2018) to analyze whether known biases in NLI data have been removed in the trained sentence representations. In particular, following Belinkov et al. (2019b), we train a classifier which only sees the representation of the hypothesis sentence and see if it can predict the class of the sentence pair, which is an established criterion to measure known biases in NLI datasets (Gururangan et al., 2018). Thus, we freeze the trained encoders from our model and the BERT baseline and retrain a hypothesis-only classifier on hypotheses from the SNLI and MNLI datasets.[6] For reference, we compare to a hypothesis-only model with a BERT encoder trained end-to-end. Table 4 shows the results. With the baseline (BERT), the retrained classifier is not able to recapture all the biases (H-only), but it captures much more than with our method (VIBERT). VIBERT is so successful at reducing biases that performance of the hypothesis-only classifier is close to chance (33%).

Table 4: Hypothesis-only accuracy when freezing the encoder from models trained on SNLI/MNLI in Table 2 and retraining a hypothesis-only classifier (BERT, VIBERT), and baseline results when the encoder is not frozen (H-only). Lower results show more successful debiasing.

| Model | SNLI | | | MNLI | | |
|---|---|---|---|---|---|---|
| | **Train** | **Dev** | **Test** | **Train** | **Dev** | **Test** |
| H-only | 81.3 | 61.89 | 62.17 | 87.15 | 53.46 | 53.63 |
| BERT | 66.40 | 53.73 | 53.17 | 58.5 | 44.68 | 44.03 |
| VIBERT | 38.20 | **36.65** | **37.10** | 42.03 | **36.43** | **35.75** |

**Impact of VIB on Overfitting**   To analyze the effect of VIB on reducing overfitting, we analyze the effect of the $\beta$ parameter on training and validation error since $\beta$ controls the trade-off between removing information from the sentence embedding (high $\beta$) and keeping information that is predictive of the output (low $\beta$). We fix the bottleneck size ($K$) based on the models selected in Section 3.1, and we train VIBERT on the GLUE benchmark for varying values of $\beta$ and plot the validation and training loss in Figure 3.

For small values of $\beta$, where VIB has little effect, the validation loss is substantially higher than the training loss, indicating overfitting. This is because the network learns to be more deterministic ($\Sigma \approx 0$), thereby retaining too much irrelevant information. As we increase $\beta$, where VIB has an effect, we observe better generalization performance with less overfitting. As $\beta$ becomes too large, both the training and validation losses shoot up because the amount of preserved information is insufficient to differentiate between the classes. This pattern is observable in the MRPC and RTE datasets, with a similar pattern in the STS-B dataset.

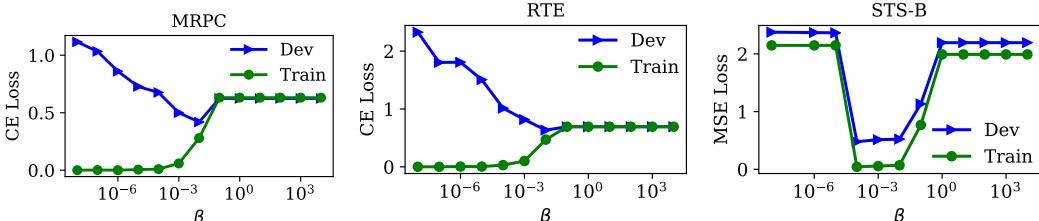

Figure 3: Validation and training losses of VIBERT for varying $\beta$ and a fixed bottleneck size on GLUE.

**Efficiency Evaluation**   Table 5 presents the efficiency evaluation in terms of memory, number of parameters, and time for all the methods measured on RTE. Our approach has several attractive properties. First, while our method is slightly larger in terms of parameters compared to the other standard regularization approaches due to an additional MLP layer (Figure 1), the difference is still marginal, and for BERT$_{Base}$ model with 109.48M trainable parameters, that is less than 1.22% more parameters. Second,

---

[6]Note that with VIBERT, the frozen encoder $p_\theta(z|x)$ outputs a distribution, and the hypothesis-only classifier is trained on samples from this distribution.

Table 5: Performance evaluation for all methods. $\Delta\%$ are relative differences with BERT.

| Model | Memory | $\Delta\%$ | #Parameters | $\Delta\%$ | Time | $\Delta\%$ |
|---|---|---|---|---|---|---|
| BERT | 290.91 GB | — | 109.48 M | — | 4.50 min | — |
| +Mixout | 407.65 GB | 40.13 % | 109.48 M | 0% | 5.15 min | 14.44% |
| +WD | 331.78 GB | 14.05% | 109.48 M | 0% | 4.91 min | 9.11% |
| +Dropout | 290.91 GB | 0% | 109.48 M | 0% | 4.68 min | 4% |
| VIBERT | 292.57 GB | 0.57 % | 110.83 M | 1.22% | 4.67 min | 3.77% |

our approach presents a much better memory usage with low-overhead, close to Dropout, while WD and especially Mixout cause substantial memory overhead. In dealing with large-scale transformer models like BERT, efficient memory usage is of paramount importance. Third, in terms of training time, our method is similar to Dropout and much faster than the other two baselines. Relative to BERT, VIBERT increases the training time by 3.77%, while WD and Mixout cause the substantial training overhead of 9.11% and 14.44%. Note that our method and other baselines require hyper-parameter tuning.

**Ablation Study**    As an ablation, Table 6 shows results for our model without the compression loss (VIBERT ($\beta = 0$)), in which case there is no incentive to introduce noise, and the VIB layer reduces to deterministic dimensionality reduction with an MLP. We optimize the dimensionality of the MLP layer ($K$) as a hyper-parameter for both methods. This ablation does reduce performance on all considered datasets, demonstrating the added benefit of the compression loss of VIBERT.

Table 6: Average ablation results over 3 runs with std in parentheses on GLUE. BERT and VIBERT's results are from Table 1.

| Model | MRPC | | STS-B | | RTE |
|---|---|---|---|---|---|
| | Accuracy | F1 | Pearson | Spearman | Accuracy |
| BERT | 87.80 (0.5) | 83.20 (0.6) | 84.93 (0.1) | 83.53 (0.0) | 67.93 (1.5) |
| VIBERT ($\beta$=0) | 88.57 (0.6) | 84.27 (0.7) | 87.10 (0.4) | 86.00 (0.5) | 69.63 (1.3) |
| VIBERT | **89.23** (0.1) | **85.23** (0.2) | **87.63** (0.3) | **86.50** (0.4) | **70.53** (0.5) |

## 5    RELATED WORK

**Low-resource Setting**    Recently, developing methods for low-resource NLP has gained attention (Cherry et al., 2019). Prior work has investigated improving on low-resource datasets by injecting large unlabeled in-domain data and pretraining a unigram document model using a variational autoencoder and use its internal representations as features for downstream tasks (Gururangan et al., 2019). Other approaches propose injecting a million-scale previously collected phrasal paraphrase relations (Arase & Tsujii, 2019) and data augmentation for translation task (Fadaee et al., 2017). Due to relying on the additional source and in-domain corpus, such techniques are not directly comparable to our model.

**Information Bottleneck**    IB has recently been adopted in NLP in applications such as parsing (Li & Eisner, 2019), and summarization (West et al., 2019). Voita et al. (2019) use the mutual information to study how token representations evolve across layers of a Transformer model (Vaswani et al., 2017). This paper – to the best of our knowledge – is the first attempt to study VIB as a regularization technique to improve the fine-tuning of large-scale language models on low-resource scenarios.

**Regularization Techniques for Fine-tuning Language models**    In addition to references given throughout, Phang et al. (2018) proposed to perform an extra data-rich intermediate supervised task pretraining followed by fine-tuning on the target task. They showed that their method leads to improved fine-tuning performance on the GLUE benchmark. However, their method requires pretraining with a large intermediate task. In contrast, our goal is to use only the provided low-resource target datasets.

## 6 Conclusion and Future Directions

We propose VIBERT, an effective model to reduce overfitting when fine-tuning large-scale pretrained language models on low-resource datasets. By leveraging a VIB objective, VIBERT finds the simplest sentence embedding, predictive of the target labels, while removing task-irrelevant and redundant information. Our approach is model agnostic, simple to implement, and highly effective. Extensive experiments and analyses show that our method substantially improves transfer performance in low-resource scenarios. We demonstrate our obtained sentence embeddings are robust to biases and our model results in a substantially better generalization to out-of-domain NLI datasets. Future work includes exploring incorporating VIB on multiple layers of pretrained language models and using it to jointly learn relevant features and relevant layers.

## Acknowledgements

We would like to thank Maksym Andriushchenko for his helpful comments. Rabeeh Karimi was supported by the Swiss National Science Foundation under the project Learning Representations of Abstraction for Opinion Summarisation (LAOS), grant number "FNS-30216". Yonatan Belinkov was supported by the ISRAEL SCIENCE FOUNDATION (grant No. 448/20).

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

## A  EXPERIMENTAL DETAILS

**Datasets Statistics**    Table 7 shows the statistics of the datasets used in our experiments.

Table 7: Datasets used in our experiments.

| Dataset | #Labels | Train | Val. | Test |
|---------|---------|-------|------|------|
| **Single-Sentence Tasks** | | | | |
| IMDB | 2 | 20K | 5K | 25K |
| YELP | 5 | 62.5K | 7.8K | 8.7K |
| **Inference Tasks** | | | | |
| SNLI | 3 | 550K | 10K | 10K |
| MNLI | 3 | 393K | 9.8K | 9.8K |
| RTE | 2 | 2.5K | 0.08K | 3K |
| **Similarity and Paraphrase Tasks** | | | | |
| MRPC | 2 | 3.7K | 0.4K | 1.7K |
| STS-B | 1 (Similarity score) | 5.8K | 1.5K | 1.4K |

**Computing Infrastructure**    We run all experiments on one GTX1080Ti GPU with 11 GB of RAM.

**VIBERT Architecture**    The MLP module used to compute the compressed sentence representations (Figure 1) is a shallow MLP with $768$, $\frac{2304+K}{4}$, $\frac{768+K}{2}$ hidden units with a ReLU non-linearity, where $K$ is the bottleneck size. Following Alemi et al. (2017), we average over 5 posterior samples, i.e., we compute $p(y|x) = \frac{1}{5}\Sigma_{i=1}^{5}q_{\phi}(y|z_i)$, where $z^i \sim p_{\theta}(z|x)$. Similar to Bowman et al. (2016), we use a linear annealing schedule for $\beta$ and set it as $\min(1, \text{epoch} \times \beta_0)$ in each epoch, where $\beta_0$ is the initial value.

## B  HYPER-PARAMETERS

**The GLUE Benchmark Experiment**    Results on GLUE benchmark are reported in Table 1. We fine-tune all the models for 6 epochs to allow them to converge. We use early stopping for all models by choosing the model performing the best on the validation set with the evaluation criterion of average F1 and accuracy for MRPC, accuracy for RTE, and average Pearson and Spearman correlations for STS-B. For VIBERT, we sweep $\beta$ over $\{10^{-4}, 10^{-5}, 10^{-6}\}$ and $K$ over $\{144, 192, 288, 384\}$. For dropout, we use dropping probabilities of $\{0.25, 0.45, 0.65, 0.85\}$. For Mixout, we consider mixout probability of $\{0.1, 0.2, 0.3, 0.4, 0.5, 0.6, 0.7, 0.8, 0.9\}$. For WD, we consider weight decay of $\{10^{-6}, 10^{-5}, 10^{-4}, 10^{-3}, 10^{-2}, 10^{-1}, 1\}$.

**Varying-resource Experiment**    Results on varying sizes of training data are reported in Table 2. We fine-tune all models for 25 epochs to allow them to converge. We use early stopping for all models based on the performance on the validation set. We also perform hyper-parameter tuning on the validation set. Since we consider datasets of a different number of training samples, we need to account for a suitable range of bottleneck size and we sweep $K$ over $\{12, 18, 24, 36, 48, 72, 96, 144, 192, 288, 384\}$ and $\beta$ over $\{10^{-4}, 10^{-5}\}$. For dropout, we consider dropping probabilities of $\{0.25, 0.45, 0.65, 0.85\}$. For Mixout, we consider mixout probability of $\{0.1, 0.2, 0.3, 0.4, 0.5, 0.6, 0.7, 0.8, 0.9\}$. For WD, we consider weight decay of $\{10^{-6}, 10^{-5}, 10^{-4}, 10^{-3}, 10^{-2}, 10^{-1}, 1\}$.

**Ablation Experiment**    Ablation results are shown in Table 6. For VIBERT ($\beta$=0), we sweep $K$ over the same range of values as VIBERT, i.e., $\{144, 192, 288, 384\}$

## C  MAPPING

We train all models on SNLI or MNLI datasets and evaluate their performance on other target datasets. The SNLI and MNLI datasets contain three labels of contradiction, neutral, and entailment. However,

some of the considered target datasets have only two labels, such as DPR or SciTail. When the target dataset has two labels of *entailed* and *not-entailed*, as in DPR, we consider the predicted contradiction and neutral labels as the not-entailed label. In the case the target dataset has two labels of *entailment* and *neutral*, as in SciTail, we consider the predicted contradiction label as neutral.

