# OpenReview forum: "Variational Information Bottleneck for Effective Low-Resource Fine-Tuning"
_ICLR.cc/2021/Conference — ICLR 2021 Poster_

### Official Review · AnonReviewer3 · 2020-10-25
**Recommendation to reject due to lack of novelty and exclusively empirical contributions.**

**Rating:** 4
**Confidence:** 4

**Review:**

**Short summary of the paper**:
The authors apply the Deep Variational Information Bottleneck (DVIB) to a NLP setting, using pretrained BERT
as a fixed part of the encoder and fine-tune subsequent MLP layers of the encoder as well as an MLP decoder.
The proposed architecture shows state-of-the-art results compared to other recent regularization methods, especially in low-resource and out-of-domain benchmarks.

**Contributions**:
- Proposal of the use of DVIB with large-scale pretrained models such as BERT in a NLP setting (low significance)
- Extensive experiments showing higher generalization and robustness to bias compared to other SOTA regularization methods in NLI benchmarks and low-resource transfer learning (medium significance)

**Pros**:
- The shown results show SOTA results in terms of generalization for a wide range of benchmarks with only marginal increase of model complexity (in terms of # of parameters & training-time).


**Cons**:
Limited novelty & incremental contribution:
- Although SOTA results are shown in very extensive experiments, the methodical contribution is rather marginal,
as it boils down to adding a pre-trained BERT to the encoder part of a DVIB.
- Aside from the pre-trained BERT part, no contributions or changes to a vanilla DVIB architecture were made.
- The novelty mainly stems from applying the DVIB to a new specific setting ("fine-tuning large-scale language models on low-resource scenarios").

**Style**:
Overall, the paper is well written and structured.

**Experiments**:
In principal, the experimental setup seems well reasoned, comprehensible & extensive.
However, I'm rather concerned about the general concept of "fine-tuning across random seeds".
In my opinion the random seed should not be a tunable hyperparameter.

**Minor Comments**:
I think for the effect of the Lagrange parameter on the losses (Figure 3), an IB curve plotting the two mutual information terms against each other for different betas would be more suitable.

---

> ### Author Response · Authors · 2020-11-17
> **Clarifications**
>
> We thank the reviewer for their helpful comments.  We will try to clarify these points in any future version of our paper.
>
> Reviewer comment:
> "the methodical contribution is rather marginal, as it boils down to adding a pre-trained BERT to the encoder part of a DVIB."
>
> This portrayal of our contribution is false.  We never compare models with BERT to models without BERT, and we use the scientific method to evaluate all our contributions.  Also, as discussed in the general comments above, we do not claim a novel contribution to VIB itself; our contribution is in identifying the ways in which VIB is useful for fine-tuning.  Please review the paper which we actually submitted.
>
> Reviewer comment:
> "In my opinion the random seed should not be a tunable hyperparameter."
>
> We'd like to clarify a misunderstanding: We did not tune on the random seeds. In tables 1, 2, and 6, where we deal with low-resource datasets, we trained all methods for 3 random seeds, and report the averaged results and the std across all three seeds.
>
> But tuning the random seed is a training methodology which many developers use in practice.  In Figure 2, we followed the methodology proposed in Dodge et al, 2019, whose goal is to study how much random seeds impact the results. For this, we considered the selected VIBERT model and baseline, train both models for 50 random seeds, and plot the expected test performance when selecting the best seed from a budget of X random seeds. The results on Figure 2 demonstrate that our VIBERT model consistently obtains better performance than BERT on all datasets, and makes it very clear that our higher performance is not a result of the choice of random seeds.
>
> Reviewer comment:
> "I think for the effect of the Lagrange parameter on the losses (Figure 3), an IB curve plotting the two mutual information terms against each other for different betas would be more suitable."
>
> This would also make sense, but it would make no difference to the conclusion.  CE-Loss and I(Z,Y) are monotonically related, and Beta and I(Z,X) are monotonically related.  We chose the version which we thought would be more easily understood by most readers to show the impact of reducing over-fitting.

---

> > ### Comment · AnonReviewer3 · 2020-11-23
> > **RE: Clarifications**
> >
> > Thank you for the response, especially regarding the random seeds.
> > Regarding the IB Curve, it does make sense to just use the CE-Loss in context of targeting a more broad audience.
> >
> > Regarding the novelty:
> >
> > I acknowledge that the proposed method of using a VIB in this specific setting has not been done before, and as you noted, the paper never claimed a novel contribution to the VIB itself.
> >
> > However, I stand by my opinion that it is limited in its novelty, as I would expect either more methodical or more formal  contributions aside from empirically showcasing improvements in terms of accuracy and robustness in a specific setting.
> >
> > (As a Side Note: The robustness was mentioned in the Contributions part of my review).

---

### Official Review · AnonReviewer4 · 2020-10-28
**Well written paper with good results, but limited novelty**

**Rating:** 4
**Confidence:** 4

**Review:**

This paper studies fine-tuning BERT-like pretrained language models (PLMs) on low resource target tasks. The authors hypothesize that the general-purpose knowledge obtained by the PLMs from pre-training might be irrelevant and redundant for a given target task. When fine-tuned onto a low resource target task, overfitting is likely to happen. To this end, a fine-tuning framework based on variational information bottleneck (VIB) is proposed to address these challenges. Specifically, the sentence representation will be mapped to a latent Gaussian variable  which compresses information in the sentence and also suppress irrelevant and redundant features, and a reconstructed version of the representation is used for task prediction. Empirical evaluations on sever datasets demonstrates the effectiveness of the method over previous research.

The paper is presented well, and it's a good read. However, my major concern is on the novelty of the proposed method. As cited by the paper, VIB has been proposed and explored in various different settings, including supervised learning, semi-supervised learning, etc., and in a similar sense, variational encoder decoders have also been thoroughly explored. The proposed method is a direct application of VIB and/or variational encoder decoder. Apart from the competitive experimental results shown on the GLUE benchmark and a set of other tasks over standard baselines including Dropout, mixout and weight decay, I find it hard to justify the novelty of the proposed method. In other words, the VIB framework is general and if additional task/fine-tuning specific insights were identified and shown to necessary when applying to the low-resource fine-tuning, novelty is also justified. However, with the current set up of plainly applying VIB to fine-tune a PLM, I find novelty rather limited.

A minor question: as hypothesized if the pretrained LM contains many general purpose features, thus those irrelevant and redundant features needs to be suppressed, would one imagine the framework to work even better with large pretrained model pretrained on a much larger corpus (like BERT-large compared to BERT-base)? The main results in the paper seem to suggest otherwise, i.e., with a larger model, VIBERT actually has much less room to improve. How would the VIB framework work with a different PLM, e.g., XLM-Roberta, XLNet or T5?

---

> ### Author Response · Authors · 2020-11-17
> **Clarifications**
>
> We thank the reviewer for their helpful comments.  We will try to clarify these points in any future version of our paper.
>
> Reviewer comment:
> "the VIB framework is general and if additional task/fine-tuning specific insights were identified and shown to necessary when applying to the low-resource fine-tuning, novelty is also justified. However, with the current set up of plainly applying VIB to fine-tune a PLM, I find novelty rather limited."
>
> As discussed in the general comments, we do not claim a novel contribution to VIB itself; our contribution is in identifying the ways in which VIB is useful for fine-tuning.  Please see our comment above.
>
> Reviewer comment:
> "would one imagine the framework to work even better with large pretrained model pretrained on a much larger corpus (like BERT-large compared to BERT-base)? The main results in the paper seem to suggest otherwise,"
>
> This is a good question, but we leave it for future work.  Our guess is that this is due to the difficulty of optimization with BERT_large and VIBERT_large, as suggested by the higher variance of results with BERT_large and VIBERT_large compared to BERT_base and VIBERT_base.  We note that in both settings, our method obtains higher performance compared to the previous state-of-the-art regularization techniques like Mixout for finetuning large-scale language models.
>
> Reviewer question:
> "How would the VIB framework work with a different PLM, e.g., XLM-Roberta, XLNet or T5?"
>
> Thank you, this is a good suggestion for future work. However, as confirmed by reviewers, this paper already includes extensive experiments.  We do our experiments with BERT because it is the most widely used pretrained model for finetuning.  For this reason, we consider experimenting with other language models to be out of the scope of this paper.

---

### Official Review · AnonReviewer2 · 2020-10-29
**Nice work on using information bottleneck for fine-tuning pre-trained models**

**Rating:** 8
**Confidence:** 4

**Review:**

This work applies information bottleneck as a way to compress the pre-trained representation so that only meaningful features are employed for the target task. It is applied for the number of GLUE tasks especially focusing on low resource settings and show consistent gains over previously known strong baselines, e.g., Mixout and L2-of-difference. This work also demonstrates that the learned model has generalization capacity so that the tuned model works on out-of-domain data.

# Pros

* An elegant solution to the fine tuning settings especially for the low-resource settings.

* Experiments are performed extensively on various tasks and demonstrates its effectiveness in generalization for out-of-domain settings.

* Interesting analysis of the experimental results.

# Cons

* Basic idea is already demonstrated by Li and Eisner (2019), and I was not very surprised by this results.

# Details

It is a very sophisticated way of avoiding overfitting especially when the data size is limited, and it might have an impact of broader application when exploiting pre-trained models. Thus, I'd recommend acceptance for this submission.

---

> ### Author Response · Authors · 2020-11-17
> **Clarification**
>
> We thank the reviewer for their helpful comments.
>
> Reviewer comment:
> "Basic idea is already demonstrated by Li and Eisner (2019)"
>
> Li and Eisner(2019) use VIB to learn to compress contextualized word embeddings for parsing tasks.  They do not consider the low-resource scenario nor fine-tuning, and do not address any of the three contributions we claim in our paper.

---

### Official Review · AnonReviewer1 · 2020-11-01
**The paper proposes a method to avoid overfitting while finetuning the large pretrained models for downstream tasks on small scale datasets.**

**Rating:** 7
**Confidence:** 3

**Review:**

The paper proposes a method to avoid overfitting while finetuning the large pretrained models for downstream tasks on small scale datasets. It has been shown that many SOTA models usually overfit w.r.t. spurious correlations in the data and as a result fail miserably when tested for generalization on the out of domain datasets. The proposed method tries to maximally filter out task-irrelevant information in the feature vectors by minimizing the mutual information between the original features and the bottleneck features while simultaneously optimizing for performance. Experiments on several datasets show improved performance on both in-domain and out-of-domain datasets.

Strong Points:
Simple to implement the method and strong empirical results and analysis. In Section 3, Table 2 clearly shows that the method provides significant improvements under the low-data regimes and the model also achieves significant improvements in most of the datasets when tested for out of domain generalization. Analysis in section 4 shows that the method is indeed able to avoid overfitting to spurious correlations.

Weak Points:
Since the paper deals with low-resource scenarios, I would have really appreciated if the experiment section also included some experiments on multilingual datasets while focusing on low-resource languages.  The general usefulness of the proposed method might have been more apparent if the paper could also cover a few additional tasks beyond text classification (E.g. NER, Translation e.t.c.)

Questions:
In Section 3.3, the authors methon the following:
> Following prior work (Belinkov et al., 2019a;Mahabadi et al., 2020), we select hyper-parameters based on the development set of each target dataset and report the results on the test set.

Where is the hyperparameter selection required while evaluating the model on OOD data? I guess I am missing something here...

---

> ### Author Response · Authors · 2020-11-17
> **Answers to questions**
>
> We thank the reviewer for their helpful comments.
>
> Reviewer question:
> "Questions: In Section 3.3, the authors methon the following:
> > Following prior work (Belinkov et al., 2019a;Mahabadi et al., 2020), we select hyper-parameters based on the development set of each target dataset and report the results on the test set.
>
> Where is the hyperparameter selection required while evaluating the model on OOD data? I guess I am missing something here..."
>
> Similar to other regularization techniques, VIB also requires hyper-parameters to define the strength of regularization. As explained in Section 2, below equation (2), K and \beta are hyper-parameters for the VIB method, specifying the strength of regularization.  The reason hyperparameter selection needs to be done on OOD data and not on in-domain data is that the strength of regularization needs to take into consideration how similar the training data is to the OOD testing data.
>
> Reviewer comment:
> "Since the paper deals with low-resource scenarios, I would have really appreciated if the experiment section also included some experiments on multilingual datasets while focusing on low-resource languages. The general usefulness of the proposed method might have been more apparent if the paper could also cover a few additional tasks beyond text classification (E.g. NER, Translation e.t.c.)"
>
> Thank you, this is a nice suggestion, and it would be good for future work. However, as confirmed by reviewers, this paper already includes extensive experiments on various tasks and is already heavy on experimental sections, so we consider including additional experiments out of scope of this submission.

---

### Author Response · Authors · 2020-11-17
**Simplicity does not imply a lack of novelty if it is not at all obvious**

Two of four reviewers claim a lack of novelty.  It is true that we are not proposing a novel version of VIB; VIB is just a tool for us.  Our novel contributions are the proposal and evaluation of VIB as a method for avoiding both overfitting and dataset biases during fine-tuning.  VIB has never been proposed for fine-tuning, and VIB has never been proposed as a bias reduction method.  So clearly these contributions are novel, and the only remaining question is whether they are too obvious to be considered a contribution.

The use of VIB for fine-tuning is not at all obvious.  The BERT paper currently has over 12,000 citations.  Most of those are doing fine-tuning, but none of them use VIB for fine-tuning.  Some of this work is in a low-resource setting, where we have shown VIB is clearly effective.  Why have hundreds of top-level researchers never done this before?  It must not be at all obvious.  Several papers (see Baselines in Section 3) have specifically looked at overfitting during fine-tuning, but none of those have used VIB either.

The use of VIB for bias reduction is not at all obvious.  Reducing susceptibility to dataset biases is an active area of research in NLP (see references in Section 3.3), but none of this work has used VIB.  We think that the conclusion that VIB can distinguish superficial dataset correlations from deep semantic correlations is an extremely exciting development in this sub-field.  One of these reviews does not even mention the contribution of robustness to dataset biases (contribution 3) as a contribution.

---

### Decision · Program_Chairs · 2021-01-07
**Final Decision**

**Decision:**

Accept (Poster)

**Comment:**

The paper shows the success of a relatively simple idea -- fine tune a pretrained BERT Model using Variational Information Bottleneck method of Alemi to improve transfer learning in low resource scenarios.

I agree with the reviewers that novelty is low -- one would like to use any applicable method for controlling overfitting when doing transfer learning, and of the suite of good candidates, VIB is an obvious one -- but at the same time, I'm moved by the results because of: the improvements and the success on a wide range of tasks and the surprising success of VIB over other alternatives like dropout etc, and hence I'm breaking the tie in the reviews by supporting acceptance.  Its a nice trick that the community could use, if the results of the paper are an indication of its potential.